# Structure-Function Relationship in Patients with Retinitis Pigmentosa and Hyperautofluorescent Rings

**DOI:** 10.3390/jcm11175137

**Published:** 2022-08-31

**Authors:** Soung Jun Kim, Chae Hyun Song, Kun Ho Bae, Chang Ki Yoon, Un Chul Park, Eun Kyoung Lee

**Affiliations:** 1Pre-Medical Program, Seoul National University College of Medicine, Seoul 03080, Korea; 2Department of Ophthalmology, Seoul National University College of Medicine, Seoul National University Hospital, Seoul 03080, Korea

**Keywords:** fundus autofluorescence, hyperautofluorescent ring, optical coherence tomography, outer retinal thickness, retinitis pigmentosa, segmented linear regression

## Abstract

This study aimed to investigate the association between retinal sensitivity and retinal microstructures in fundus autofluorescence (FAF) and optical coherence tomography (OCT) in patients with retinitis pigmentosa (RP) and hyperautofluorescent (hyperAF) rings. This cross-sectional study included 44 eyes from 26 consecutive patients with RP. The morphological geometry of the hyperAF ring, such as three distinct FAF regions, hyperAF ring area, and longest diameter of the hyperAF ring on FAF, and the retinal microstructure, such as total retinal thickness (TRT) and outer retinal thickness (ORT), on OCT, were evaluated. A strong correlation of mean retinal sensitivity with hyperAF ring area (R = 0.8013, *p* < 0.001) and longest diameter of the hyperAF ring (R = 0.9072, *p* < 0.001) was observed. Segmented linear regression (SLR) analysis revealed breakpoints of 12.83 mm^2^ and 5.21 mm, respectively. ORT (R = 0.6551, *p* < 0.001) was more strongly correlated with retinal sensitivity than TRT (R = 0.2732, *p* < 0.001). SLR analysis revealed a breakpoint between the ORT and retinal sensitivity of 145.12 μm. In patients with RP and hyperAF rings, retinal sensitivity was strongly associated with the morphological geometry of the hyperAF ring. ORT, rather than TRT, strongly correlated with retinal sensitivity.

## 1. Introduction

Retinitis pigmentosa (RP) is an inherited retinal disorder characterized by the primary degeneration of rod and cone photoreceptors. More than 1.5 million individuals are affected globally, with a prevalence of approximately 1:4000 [1]. The disease is genetically heterogeneous, and more than 80 mutated genes have been identified [2]. RP can be inherited in autosomal dominant (30–40%), autosomal recessive (50–60%), or X-linked (5–15%) manner [3]. Patients present with night blindness (nyctalopia), visual field constriction, and an eventual decrease in a central vision leading to blindness in later stages [4]. In patients with early-stage RP, central vision often remains intact until later in the disease [3]. Many patients with RP who maintain central vision are aware of subjective visual impairment; however, vision itself cannot represent the patient’s perception of visual impairment.

Fundus autofluorescence (FAF) is a non-invasive technique that allows visualization of the distribution of lipofuscin and demonstrates the health and metabolic function of the retinal pigment epithelium (RPE) [5]. The presence of hyperautofluorescent (hyperAF) rings in patients with RP has been reported in 59–94% of patients [6,7,8] and may represent an abnormal perifoveal accumulation of lipofuscin in the RPE attributable to increased outer segment dysgenesis as a precursor of apoptosis in RP [9,10]. Previous studies have shown that the site of the hyperAF ring correlates with a zone of impaired integrity of the photoreceptor layer on optical coherence tomography (OCT), and patients with a hyperAF ring had significantly better visual acuity or mean deviation measured with a Humphrey perimeter than those with foveal hyperautofluorescence [6,10]. Therefore, the structural and functional implications of the hyperAF ring in RP patients and its prognostic and monitoring value have been well established [6,7,11,12,13]. However, there is still a need for a more robust correlation between structural abnormalities and visual functional impairments in patients with RP and a hyperAF ring.

Fundus-related microperimetry allows simultaneous imaging of the fundus while projecting light stimuli onto a testing point when assessing macular function. Precise topographical evaluation of retinal sensitivity at a specific point enables the generation of a retinal sensitivity map registered with a fundus image. Few reports have examined the correlation between retinal structure assessed by FAF and OCT and retinal function assessed quantitatively using a microperimetry device. This study evaluated the correlation of retinal microstructures on FAF and OCT with retinal sensitivity in patients with RP and hyperAF rings. 

## 2. Materials and Methods

### 2.1. Participants

This study was performed at Seoul National University Hospital in Korea. The study adhered to the Declaration of Helsinki and was approved by the Institutional Review Board of Seoul National University Hospital (IRB approval number: 2004-009-1115). We retrospectively reviewed the medical records of patients diagnosed with RP who visited the Inherited Retinal Disease Clinic at Seoul National University Hospital between March and December 2020. The clinical diagnosis of RP was based on presenting symptoms, family history, complete ophthalmic examination, imaging, visual field examination, and full-field electroretinography (ERG) testing. We only included patients with RP who exhibited a hyperAF ring on FAF imaging. The exclusion criteria were as follows: (1) complications such as cystoid macular edema or epiretinal membrane with macular traction on OCT images; (2) a coexisting ocular pathology that could potentially impair visual function (e.g., comorbid maculopathy); (3) optical media opacity that could significantly interfere with image acquisition (e.g., cataract of more than Emery-Little classification grade III); and (4) poor cooperation with the microperimetry examination such as unstable fixation.

### 2.2. Ocular Examination

All patients underwent a comprehensive ophthalmologic examination, including measurement of best-corrected visual acuity (BCVA), slit-lamp biomicroscopy, dilated fundus examination, ultra-widefield fundus photography, FAF, spectral domain OCT (SD-OCT), visual field examination, microperimetry, and full-field ERG. Fundus examinations were conducted using a Vx-10 (Kowa OptiMed, Tokyo, Japan) and Optos 200Tx (Optos PLC, Scotland, UK). FAF images were obtained using Spectralis (Heidelberg Retina Angiograph; Heidelberg Engineering, Heidelberg, Germany), and SD-OCT was conducted using Spectralis and Zeiss Cirrus 4000 (Carl Zeiss Meditec, Dublin, CA, USA). A visual field examination was conducted using a Humphrey Field Analyzer II (Carl Zeiss Meditec) and Goldmann manual perimeter (Haag-Streit, Berne, Switzerland). Full-field ERG was performed using gold foil recording electrodes and incorporated the ISCEV (International Society for Clinical Electrophysiology of Vision) standard protocols [14]. The BCVA measurements were converted to logarithm of minimal angle of resolution (logMAR) units for statistical analysis.

### 2.3. Microperimetry

Retinal sensitivity measurements were performed using macular integrity assessment (MAIA) microperimetry (CentreVue, Padova, Italy) under mesopic conditions without dark adaptation. This device uses Goldmann III-sized stimuli with a dynamic range of 0–36 dB and maximum luminance of 318.3 cd/m^2^ on a background of 1.27 cd/m^2^. Real-time fundus imaging and eye tracking provided a secure correspondence of light stimulus and retinal location during the examination. The standard expert test included 37 points (4−2 strategy). It covered the central 10° diameter circle, with retinal sensitivities at 1°, 3°, and 5° radii (12 stimuli at each degree and one stimulus at the central fovea), respectively (Figure 1A). At the end of each examination, the following results were exported: threshold retinal sensitivity (TRS) and fixation index (P1 and P2). The fixation index refers to the control of fixation losses and also registers the fixation pattern. P1 and P2 have calculated values and represent the percentage of fixation points inside a circle of 2° and 4° diameter, respectively. Fixation was considered “stable” when the P1 value was >75%, “relatively stable” when the P1 was <75%, but P2 was >75%, and “unstable” when both P1 and P2 were <75%. The fixation location was confirmed by two investigators (S.J.K. and C.H.S.). The microperimetric results of patients with unstable fixation were excluded from the analysis.

### 2.4. Association between FAF Imaging and Microperimetry

On the same day of microperimetry testing, FAF images were acquired, covering a 30° × 30° field of view centered on the fovea. An optically pumped solid-state laser (488 nm) was used for excitation, and a wide band-pass filter with a cutoff at 521 nm was used for detection. The standard procedure was followed for acquiring FAF images, including the focus of the retinal image in infrared reflection mode at 820 nm, sensitivity adjustment at 488 nm, and acquisition of images for each eye encompassing the macular area with at least a portion of the optic disc. To improve the signal-to-noise ratio, 30 images were aligned, and a mean image with 768 × 768 pixels was calculated using the scanning laser ophthalmoscope (HRA2) software (Heidelberg Explorer software version 1.10.4.0; Heidelberg Engineering, Heidelberg, Germany). The MAIA retinal sensitivity map image was superimposed on the FAF image to obtain the highest correspondence, using the retinal vessels as landmarks (Figure 1B). Retinal sensitivities were determined for each stimulus point on the MAIA for each of the three distinct FAF regions (inside the hyperAF ring, over the hyperAF ring, and outside the hyperAF ring). 

The external and internal boundaries of the hyperAF ring are defined as the visible limits on the FAF. To determine the correlation between retinal microstructures in FAF and retinal sensitivity in microperimetry, the hyperAF ring area and the longest diameter of the hyperAF ring were measured manually, and these measurements were based on the internal boundary of the hyperAF ring. We used polygon selection and a straight module in ImageJ (National Institutes of Health, Bethesda, MD, USA). HyperAF ring measurements using FAF images were independently evaluated by two investigators (S.J.K. and C.H.S.). Since the results were expressed as a pixel value, they were replaced with a micrometer (μm) value in consideration of the magnification of the FAF image. The length per pixel was calculated by counting the number of pixels equivalent to 200 μm in the FAF image using the ImageJ straight module. 

### 2.5. Association between OCT Imaging and Microperimetry

On the same day of microperimetry testing, SD-OCT images were obtained, covering a 15° × 15° field of view centered on the fovea. The MAIA retinal sensitivity map image was superimposed on the infrared image of OCT to obtain the highest correspondence using the retinal vessels as landmarks. The horizontal length of the OCT image was adjusted to be equal to the length of the superimposed image and was placed at the bottom and right sides of the superimposed image (Figure 1C). In each eye, at 37 points on the infrared OCT image corresponding to the 37 testing points by microperimetry, the retinal microstructure was evaluated for the following parameters: total retinal thickness (TRT), which was defined as the length from the internal limiting membrane (ILM) to the RPE, and outer retinal thickness (ORT), which was defined as the length from the outer plexiform layer (OPL) to the RPE. The software automatically segmented the ILM, RPE, and OPL layers. The TRT and ORT were measured manually using the built-in software in the SD-OCT system, and after that, all images were reviewed and manually corrected if incorrect segmentation was observed by a masked observer (C.K.Y.).

### 2.6. Statistical Analysis

We used the Seaborn, Numpy, and Pandas models in Google Collaboratory and IBM SPSS statistics to interpret the results. The data were organized into data frames through the Pandas model, and the mean, standard deviation, median, and quartile were identified using Numpy and IBM SPSS. Mean retinal sensitivity was compared according to the topographical distribution of the testing points on microperimetry based on the hyperAF ring on FAF images using the Seaborn model. The normality test, Kruskal–Wallis H test, and Mann–Whitney *U* test was used. We plotted the data as a scatterplot to analyze the correlation between hyperAF ring area and the longest diameter of the hyperAF ring with mean retinal sensitivity. We then conducted linear regression and segmented linear regression (SLR). Correlation coefficient (R) and *p* values were obtained using the Seaborn model. The correlation between TRT and ORT measurements and retinal sensitivity was analyzed similarly. *p* values of less than 0.05 were considered statistically significant.

## 3. Results

### 3.1. Demographics

This study included 44 eyes of 26 consecutive patients with RP and a hyperAF ring. Table 1 summarizes the patient demographics and clinical characteristics. Sixteen patients (61.5%) were men, and 10 (38.5%) were women. All patients were Korean except for one Thai patient. The mean age of the patients was 42.96 ± 14.62 years. Of the 26 enrolled patients with RP and hyperAF rings, 18 patients with both eyes were included. Only one eye of the remaining eight patients was analyzed because of poor FAF image acquisition in seven patients and unstable fixation in microperimetry in one patient. The poor FAF image quality was when the image was not taken centered on the macula, or the hyperAF ring was difficult to evaluate because the FAF image was dark. All the included patients demonstrated diffuse chorioretinal atrophy with typical bone spicule pigmentation on dilated fundus examination. There was no patient with pericentral, sectoral, or syndromic RP. The mean BCVA of the right eye was logMAR 0.11 ± 0.10, and that of the left eye was logMAR 0.09 ± 0.16, indicating that most patients maintained good central visual acuity. The central visual field (CVF) was preserved at ≤5° in 6 eyes (13.7%), 5° < CVF ≤ 10° in 18 eyes (40.9%), 10° < CVF ≤ 20° in 18 eyes (40.9%), and >20° in 2 eyes (4.5%). The mean follow-up duration was 5.13 ± 3.50 years.

### 3.2. Microperimetry Analysis

The mean macular integrity on microperimetry was 94.26 ± 16.36, and the mean fixation stability was 93.66 ± 9.20%. Forty-two eyes (95.5%) showed “stable” fixation, and two eyes (4.5%) showed “relatively stable” fixation. A total of 1628 testing points for retinal sensitivity were evaluated for their pointwise correlation with microstructural features based on FAF and SD-OCT images. The relationship between BCVA and mean retinal sensitivity was not statistically significant (*p* = 0.129).

### 3.3. Association between the HyperAF Ring and Microperimetry

Microperimetry results were assessed inside, over, and outside the hyperAF ring in each eye. At all testing points for the retinal sensitivity map superimposed on the FAF image, the topographical correlation with the hyperAF ring could be clearly determined in 1589 out of 1628 spots. The percentage of testing point distribution was 1080 (67.98%) inside the hyperAF ring, 248 (15.61%) over the hyperAF ring, and 261 (16.43%) outside the hyperAF ring. The retinal sensitivity of the testing points according to the topographical distribution of the hyperAF ring is shown in Figure 2. The mean retinal sensitivities were 24.28 ± 3.76, 17.25 ± 6.09, and 11.64 ± 7.50 dB inside the hyperAF ring, over the hyperAF ring, and outside the hyperAF ring, respectively. According to the Kruskal–Wallis H test, the mean retinal sensitivity significantly differed among the three groups (*p* < 0.001). Post hoc analysis revealed that the mean retinal sensitivity of the testing points over the hyperAF ring was greater than that outside the hyperAF ring (*p* < 0.001) but lower than that inside the hyperAF ring (*p* < 0.001). 

The correlation between the hyperAF ring area and the mean retinal sensitivity and between the longest diameter of the hyperAF ring and the mean retinal sensitivity was evaluated. Thirty-six eyes of 21 patients were included in this analysis, excluding those whose hyperAF ring boundaries were not distinctly delineated. The results are shown in Figure 3 and Figure 4, respectively. The mean area of the hyperAF ring was 8.80 ± 6.47 mm^2^, and the mean longest diameter of the hyperAF ring was 3.82 ± 1.50 mm. In two eyes of one patient included in the analysis, the area of the hyperAF ring was greater than 60 mm^2^. There was a strong positive correlation between the hyperAF ring area and mean retinal sensitivity (R = 0.8013, *p* < 0.001, Figure 3A) and a strong positive correlation between the longest diameter of the hyperAF ring and mean retinal sensitivity (R = 0.9072, *p* < 0.001, Figure 3B). SLR analysis was performed to identify breakpoints and broken-line relationships between the topographical characteristics of the hyperAF ring and retinal sensitivity. According to the SLR, there was a strong positive correlation between the hyperAF ring area and mean retinal sensitivity (R^2^ = 0.7930, *p* < 0.001, Figure 4A) and between the longest diameter of the hyperAF ring and mean retinal sensitivity (R^2^ = 0.8757, *p* < 0.001, Figure 4B). The breakpoints were 12.83 mm^2^ (95% CI [confidence interval], 12.32–13.34 mm^2^) and 5.21 mm (95% CI, 5.09–5.33 mm), respectively. Regarding the relationship between the hyperAF ring area and mean retinal sensitivity, the slope was 0.9976 (95% CI, 0.9535–1.0417) for the relationship before the breakpoint (α_1_) and was 0.0381 (95% CI, 0.0238–0.0524) for the relationship after the breakpoint (α_2_). Regarding the relationship between the longest diameter of the hyperAF ring and mean retinal sensitivity, the slope was 3.7433 (95% CI, 3.6287–3.8579) for the relationship before the breakpoint (α_1_) and was 0.2758 (95% CI, 0.1281–0.4236) for the relationship after the breakpoint (α_2_). This means that the structure-function relationship is no longer sufficient when the hyperAF ring becomes larger than a certain level in RP patients.

### 3.4. Association between Retinal Thickness and Microperimetry

A total of 520 testing points for retinal sensitivity were evaluated for their pointwise correlation with TRT and ORT measurements based on SD-OCT images. The retinal sensitivity results according to the TRT and ORT at each retinal location corresponding to the testing points of microperimetry are shown in Figure 5 and Figure 6. The mean TRT corresponding to the testing points of microperimetry was 293.44 ± 63.26 μm, and the mean ORT corresponding to the testing points of microperimetry was 158.08 ± 60.47 μm. There was a weak positive correlation between TRT and retinal sensitivity (R = 0.2732, *p* < 0.001, Figure 5A) and a moderate positive correlation between ORT and retinal sensitivity (R = 0.6551, *p* < 0.001, Figure 5B). SLR analysis was performed to identify breakpoints and broken-line relationships between retinal thickness and retinal sensitivity. When retinal sensitivity was examined in relation to TRT, no breakpoint was found, and the regression was considered significantly linear (R^2^ = 0.0746, *p* < 0.001, Figure 6A). According to the SLR, a moderate correlation was observed between the ORT and retinal sensitivity (R^2^ = 0.5178, *p* < 0.001, Figure 6B). The breakpoint between the ORT and retinal sensitivity was 145.12 μm (95% CI, 144.48–145.76 μm). Regarding the relationship between the ORT and retinal sensitivity, the slope was 0.1410 (95% CI, 0.1399–0.1421) for the relationship before the breakpoint (α_1_) and was 0.0137 (95% CI, 0.0129–0.0145) for the relationship after the breakpoint (α_2_).

## 4. Discussion

This cross-sectional study investigated the association between retinal sensitivity on microperimetry and retinal structural features on FAF and SD-OCT in patients with RP and hyperAF rings. Point-to-point analysis revealed that greater retinal sensitivity was associated with the area of and within the hyperAF ring. In line with previous reports [15], ORT was more strongly correlated with retinal sensitivity than TRT. The hyperAF ring area and the longest diameter of the hyperAF ring were positively correlated with mean retinal sensitivity. The current study is the first to suggest breakpoints between the morphological characteristics of the hyperAF ring and retinal sensitivity through SLR analysis. This study is meaningful because it confirmed the structure-function relationship through microperimetry analysis in patients with RP and hyperAF rings who maintain good central vision. It confirmed that visual acuity alone is limited in evaluating central visual function in these patients. 

Lima et al. [9] showed that the structural assessment of the retina outside the hyperAF ring revealed the absence of the inner segment/outer segment junction (IS/OS), external limiting membrane (ELM), and outer nuclear layer (ONL). At the transitional zone of the hyperAF ring, breakdown or disorganization of the IS/OS junction decreased ONL thickness in the centrifugal direction and decreased ELM detection were observed. In the region within the hyperAF ring, an intact IS/OS junction, ELM, and ONL were detected [9]. Therefore, the hyperAF ring found in RP patients represents a transition between abnormal paracentral and normal central cone function and delineates a retinal region with visual field preservation [11,16]. Lima et al. [9,13] also suggested that the hyperAF ring may reveal an anomalously high rate of photoreceptor phagocytosis and that the change in RPE function may be a secondary phenomenon due to increased metabolic load on the RPE resulting from photoreceptor apoptosis [8,9,17]. Our study demonstrated that the microstructure of the hyperAF ring evaluated by three distinct regions, the hyperAF ring area and the longest diameter of the hyperAF ring, are significantly correlated with retinal function, similar to what was observed in previous studies [7,8,11,18,19].

The SLR analysis revealed that the relationship between retinal sensitivity and the hyperAF ring area and the longest diameter of the hyperAF ring was best described with a regression function having a breakpoint between two data segments. One patient with an area of the hyperAF ring greater than 60 mm^2^ was included in the current study. This patient had a hyperAF ring extending beyond the major arcade. It is possible that microperimetry may not sufficiently reflect the retinal function of a wide area of the retina because the MAIA test covers only the central 10° diameter circle. If the size of the hyperAF ring becomes sufficiently large so that all MAIA spots exist only inside the hyperAF ring, the correlation between the hyperAF ring area and retinal sensitivity will no longer be sufficiently reflected. A ceiling effect may also have been present, so no further improvement could reasonably be expected.

Regarding retinal thickness and retinal sensitivity, we observed that ORT was more strongly correlated with retinal sensitivity than TRT. Funatsu et al. [15] investigated the retinal structure-function relationship in patients with RP by comparing microperimetry-3 images with co-registered OCT images. They showed that ORT was more strongly correlated with retinal sensitivity than TRT and suggested that these results may be due to photoreceptor cells being primarily injured in RP and that TRT can be affected by the thickness of the inner layers, which varies depending on the distance from the fovea and can be affected by the presence of an epiretinal membrane and degenerative thickening [15,20]. Although the current study minimized the variability of TRT by excluding patients with epiretinal membranes, these results indicating that ORT is more strongly correlated with retinal sensitivity than TRT are reasonable because the outer retina is primarily affected, and the inner retina is relatively preserved in patients with RP [21]. 

SLR analysis also revealed that the relationship between retinal sensitivity and ORT was best described by a regression function with a breakpoint between two data segments. This means that retinal sensitivity was relatively preserved and maintained above specific ORT measurements. Novel therapeutic approaches, such as retinal prostheses [22], cell therapy [23], gene editing therapy [24,25], and gene augmentation therapy [26], require functioning retinal structures other than photoreceptor cells. In our study, the ORT breakpoint was found to be 145.12 μm. Our results will help estimate the reference value of functioning retinal structures for novel therapeutic approaches.

The present study had several limitations. First, the study design was cross-sectional, and the sample size was small. Therefore, we did not provide information on time-dependent changes or on the identification of phenotypes associated with disease progression. Second, microperimetry was performed only once per patient. Test-retest variability and the influence of the learning effect were not reflected in the results of this study. Third, our study results could be confounded by investigator interpretation bias, although we tried to control for such bias by having two independent investigators extract data. Fourth, not performing genetic analyses of the patient cohort to demonstrate more evidence regarding the high phenotypic variability of RP patients is a limitation of our study. Finally, although we attempted to match point-to-point based on fundus images, the microperimetry testing points may not exactly overlay the raster scans of OCT. Nevertheless, the error would be less than 0.25 mm, considering the spacing between two adjacent OCT scans.

## 5. Conclusions

In conclusion, microperimetry is a useful functional tool for evaluating altered central visual function in patients with RP and hyperAF rings. Point-to-point analysis revealed that retinal sensitivity was strongly associated with the morphological geometry of the hyperAF ring. The ORT, rather than the TRT, was strongly correlated with retinal sensitivity, and SLR analysis revealed that the breakpoint between the ORT and retinal sensitivity was 145.12 μm. Further studies with larger numbers of patients are required to extrapolate our results.

## Figures and Tables

**Figure 1 jcm-11-05137-f001:**
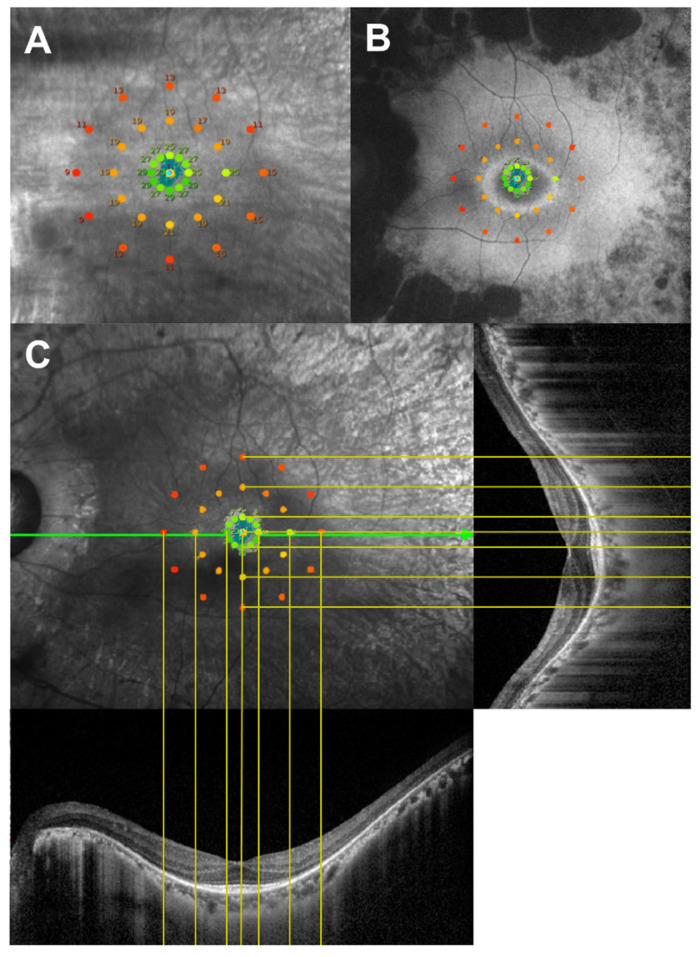
(**A**) Schematic illustration of a microperimetry sensitivity map comprising 37 testing points. (**B**) Representative superimposed image of microperimetry sensitivity map on the fundus autofluorescence image. (**C**) Optical coherence tomography images measure retinal thicknesses at the corresponding region to a stimulation point according to the superimposed translucent fundus microperimetry image.

**Figure 2 jcm-11-05137-f002:**
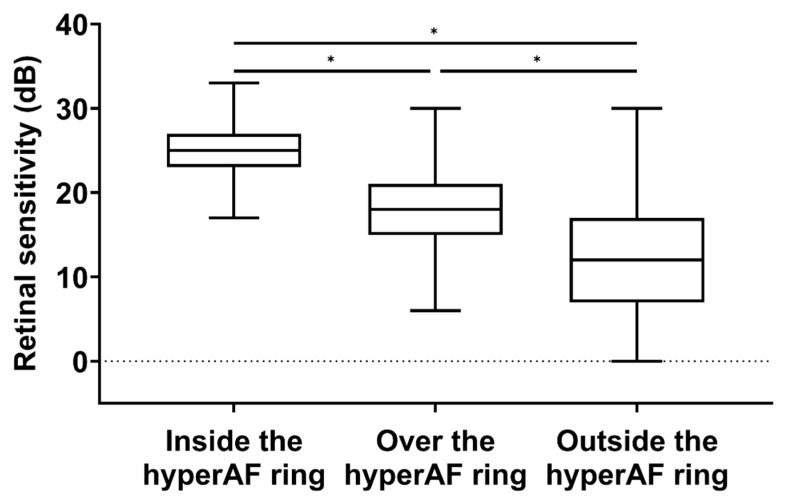
A boxplot comparing retinal sensitivity according to the topographical distribution of testing points regarding the hyperautofluorescent (hyperAF) ring. * *p* < 0.001. dB = decibels.

**Figure 3 jcm-11-05137-f003:**
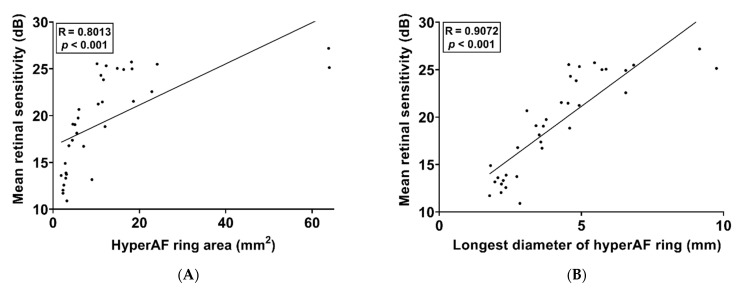
Scatterplots with linear regression analysis showing the correlation of the mean retinal sensitivity with the hyperautofluorescent (hyperAF) ring area (**A**) and the longest diameter of the hyperAF ring (**B**).

**Figure 4 jcm-11-05137-f004:**
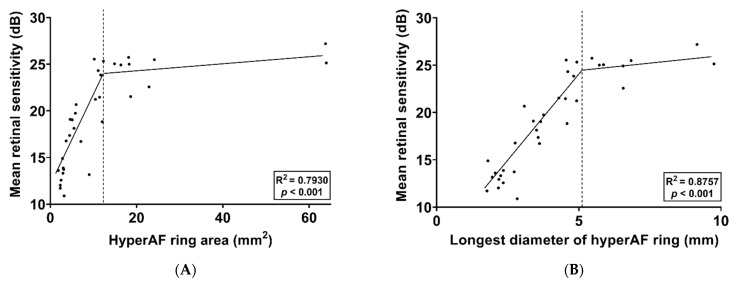
Scatterplots with segmented linear regression analysis showing the correlation of the mean retinal sensitivity with the hyperautofluorescent (hyperAF) ring area (**A**) and the longest diameter of the hyperAF ring (**B**). The breakpoints were 12.83 mm^2^ in the relationship between the hyperAF ring area and mean retinal sensitivity and 5.21 mm in the relationship between the longest diameter of the hyperAF ring and mean retinal sensitivity.

**Figure 5 jcm-11-05137-f005:**
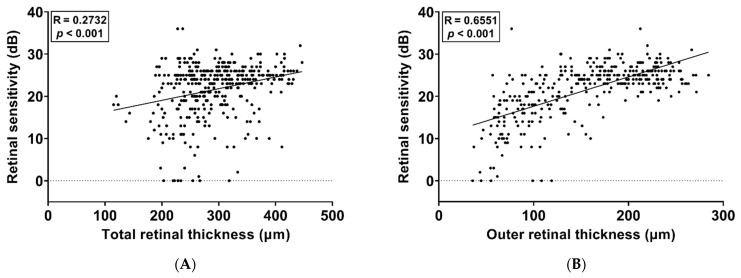
Scatterplots with linear regression analysis show the correlation of the mean retinal sensitivity with the total retinal thickness (**A**) and outer retinal thickness (**B)**.

**Figure 6 jcm-11-05137-f006:**
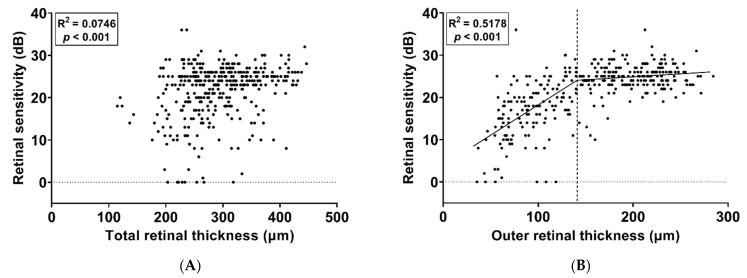
Scatterplots with segmented linear regression analysis showing the correlation of the mean retinal sensitivity with the total retinal thickness (**A**) and outer retinal thickness (**B**). No breakpoint was found, and the regression was considered significantly linear in the relationship between the total retinal thickness and mean retinal sensitivity. The breakpoint was 145.12 μm in the relationship between the outer retinal thickness and mean retinal sensitivity.

**Table 1 jcm-11-05137-t001:** Demographics and clinical characteristics of the study participants.

Variables	RP Patients with HyperAF Ring
Number of eyes (patients)	44 (26)
Male/Female	16/10
Age (years)	42.96 ± 14.62
BCVA (logMAR)	
OD	0.11 ± 0.10
OS	0.09 ± 0.16
Visual field examination	
CVF preserved at ≤5°	6
CVF preserved at 5° < CVF ≤ 10°	18
CVF preserved at 10° < CVF ≤ 20°	18
CVF preserved at >20°	2

RP = retinitis pigmentosa; hyperAF = hyperautofluorescent; BCVA = best corrected visual acuity; logMAR = logarithm of minimal angle of resolution; CVF = central visual field.

## Data Availability

The data presented in this study are available on request to the corresponding author.

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
