# Peer review of "Structure-Function Relationship in Patients with Retinitis Pigmentosa and Hyperautofluorescent Rings"

_jcm, 2022, doi:10.3390/jcm11175137_

Round 1
Reviewer 1 Report
In this interesting study, the authors investigated the structure-function relationship in patients with RP and Hyperfluorescent rings.
The manuscript is clear and generally well written.
Below are my recommendations.
METHODS
Was the RP diagnosis only clinical ? Are genetic tests available ? This could be interesting considering the high phenotypic variability of RP patients with different mutations.
The authors state that SD-OCT were acquired with Zeiss 4000 (which has a low axial resolution). Nevertheless, In figure 1 the OCT and NIR images come from a spectralis OCT. Please clarify.
RESULTS
44 eyes from 26 patients were included. Some eyes were excluded. Please specify the reason why they were excluded. The FAF quality was poor because of a cataract ? In the methods section the authors state that all eyes with Erm or CME were excluded. I suppose many eyes were excluded considering that the prevalence of CME is up to 50% of cases.
Once again there is no mention of the patients phenotype. Is there any Pericentral, Usher, RPGR, sectorial RP, etc…patient ?
Were the TRT and ORT measured on the cirrus 4000? Manually? Please specify.
DISCUSSION
The authors should acknowledge in the limitations section the lack of a genetic and phenotypic characterization.
Author Response
Reviewer comments:
Reviewer #1:
In this interesting study, the authors investigated the structure-function relationship in patients with RP and Hyperfluorescent rings.
The manuscript is clear and generally well written.
→ We appreciate this comment.
Below are my recommendations.
METHODS
Was the RP diagnosis only clinical ? Are genetic tests available ? This could be interesting considering the high phenotypic variability of RP patients with different mutations.
→ Thank you for your suggestion. The diagnosis of RP was established based on typical clinical manifestations. The current study was a retrospective study, and not performing genetic analyses of the patient cohort to demonstrate more evidence regarding the high phenotypic variability of RP patients is a limitation of our study. We have revised the limitations in the Discussion section accordingly (page 10, lines 341−342).
The authors state that SD-OCT were acquired with Zeiss 4000 (which has a low axial resolution). Nevertheless, In figure 1 the OCT and NIR images come from a spectralis OCT. Please clarify.
→ SD-OCT was conducted using two devices, Spectralis (Heidelberg Engineering, Heidelberg, Germany) and Zeiss Cirrus 4000 (Carl Zeiss Meditec, Dublin, CA, USA), which are described in the Materials and Methods section (page 2, lines 86−87).
RESULTS
44 eyes from 26 patients were included. Some eyes were excluded. Please specify the reason why they were excluded. The FAF quality was poor because of a cataract ? In the methods section the authors state that all eyes with Erm or CME were excluded. I suppose many eyes were excluded considering that the prevalence of CME is up to 50% of cases.
→ Of the 26 enrolled patients with RP and hyperAF rings, 18 patients with both eyes were included. Only one eye of the remaining eight patients was analyzed because of poor FAF image acquisition in seven patients and unstable fixation in microperimetry in one patient (page 5, lines 178−181). The poor FAF image quality was not due to a cataract, but because the image was not taken centered on the macula or the FAF image was dark, making it difficult to evaluate the hyperAF ring. We have added the above information in the Results section (page 5, lines 181−183).
Once again there is no mention of the patients phenotype. Is there any Pericentral, Usher, RPGR, sectorial RP, etc…patient ?
→ Thank you for this advice. All patients included in this study were RP patients with typical bone spicule pigmentation and diffuse chorioretinal atrophy, which are described in the Results section (page 5, lines 183−185). There was no patient with pericentral, sectoral, or syndromic RP. Since genetic testing was not performed, the phenotype of the patient according to the causative gene could not be suggested. We have revised the phenotypes of the included patients in the Results section (page 5, lines 185−186).
Were the TRT and ORT measured on the cirrus 4000? Manually? Please specify.
→ The TRT and ORT were measured manually using the built-in software in two devices of OCT (Spectralis and Zeiss Cirrus 4000). We have added the above information in the Materials and Methods section (page 4, lines 150−151).
DISCUSSION
The authors should acknowledge in the limitations section the lack of a genetic and phenotypic characterization.
→ Current study was a retrospective study, and not performing genetic analyses of the patient cohort to demonstrate more evidence regarding the high phenotypic variability of RP patients is a limitation of our study. We have revised the limitations in the Discussion section accordingly (page 10, lines 341−342).
I sincerely appreciate your hard work and effort in reviewing the manuscript. Thanks to your sincere comments, the manuscript could be a valuable and meaningful paper.

Reviewer 2 Report
This is a paper that correlates the microperimetry with OCT features like ORT and TRT. This is a cross-sectional study, so I don't understand the initial and final BVCA that is presented in table 1. It would be interesting if you compare analysis of the same patient - both eyes - and one of the eyes (RE, for example) of all patients.
Author Response
Reviewer comments:
Reviewer #2:
This is a paper that correlates the microperimetry with OCT features like ORT and TRT. This is a cross-sectional study, so I don't understand the initial and final BVCA that is presented in table 1. It would be interesting if you compare analysis of the same patient - both eyes - and one of the eyes (RE, for example) of all patients.
→ We apologize for any confusion caused by Table 1. As suggested, the current study is a cross-sectional study, and the Table 1 was revised with the BCVA at the time of FAF images and microperimetry testing (page 5, lines 186−187 & Table 1).
Of the 26 enrolled patients, 18 patients with both eyes and 8 patients with one eye were included. Since the current study investigated the structure-function relationship of the hyperAF ring for all included eyes, the analysis of comparing both eyes in one patient or monocular comparison of all patients is considered to be outside the scope of this study. Nevertheless, we appreciate this comment and look forward to planning further studies with a larger number of patients.
I sincerely appreciate your hard work and effort in reviewing the manuscript. After making your recommended revisions, this manuscript has become more robust and clearer. Thank you once again for the review.

Round 2
Reviewer 1 Report
the manuscript is now suitable for publication